# Review of Agricultural-Related Water Security in Water-Scarce Countries: Jordan Case Study

**Nabil Beithou** [1,2,*], **Ahmed Qandil** [3], **Mohammad Bani Khalid** [1], **Jelena Horvatinec** [4] **and Gabrijel Ondrasek** [4]

1   Department of Mechanical and Industrial Engineering, Applied Science Private University, P.O. Box 166, Amman 11931, Jordan; m_banikhaled@asu.edu.jo
2   Department of Mechanical Engineering, Tafila Technical University, P.O. Box 179, Tafila 66110, Jordan
3   Mechanical Engineering Department, Al Zaytoonah University of Jordan, Amman 11733, Jordan; a.qandil@zuj.edu.jo
4   Faculty of Agriculture, University of Zagreb, 10000 Zagreb, Croatia; jhorvatinec@agr.hr (J.H.); gondrasek@agr.hr (G.O.)
*   Correspondence: beithounabil@yahoo.com or n_beithou@asu.edu.jo

**Abstract:** Food security is an essential issue for human survival and civilization. Whenever food–water security is in doubt, the community is negatively affected. Globally, Jordan is the second most water-stressed country, located in an arid, politically divided and migratory active Middle East region that lacks the access to valuable natural resources such as fertile soils. Jordan receives about 78 m$^3$/person/year from renewable resources, which represents 1% of the world water share. Jordan's Water Minister declared that a 50 million m$^3$ lack of drinking water is to be faced next year; this shortage is added to the lack of irrigation water, which yields food insecurity and food price fluctuations that wear out the consumer. The aim of this study is to provide a comprehensive overview of the impact of agricultural cropping patterns and water security by analyzing the most relevant national databases. The study results will contribute to the development of national policy in order to strategize the aid programs and adaptation measures for more sustainable planning in the Jordanian agri-food sector.

**Keywords:** water management; crop production; agricultural cropping patterns; water use efficiency; water security; Jordan

## 1. Introduction

Food and water securities are major issues for nations survival and development. Significant parts of the world are struggling with water scarcity (Figure 1), including Middle East and Northern African (MENA) countries. It is estimated that two-thirds of the global world population, or over 4 billion people, is currently exposed to severe water shortages for part of the year. If climate change factors persist, a growing number of regions will suffer from water scarcity [1].

Odd distribution of water and food resources creates billions of hungry people in the world, even though one-third of the food produced is wasted [2]. This odd distribution negatively affects economic, social and environmental conditions worldwide [3]. Problems of water scarcity and food insecurity are exacerbated by climate change and increased population growth. There are needs for adaptive measures to reduce the impacts of climate change on water resources and food security [4]. The increasing global population implemented the demand for an increase in food and water resources. There are many countries in the world that are already suffering from high water scarcity, which reflects on food security by default (Figure 1). Jordan is recognized as the second-most water-stressed country in the world, located in an arid and politically divided region lacking valuable natural resources. In order to meet the increased demand for water resources, it is of great

importance to determine the strategic crops and the available arable land to save Jordan from food and water shortages.

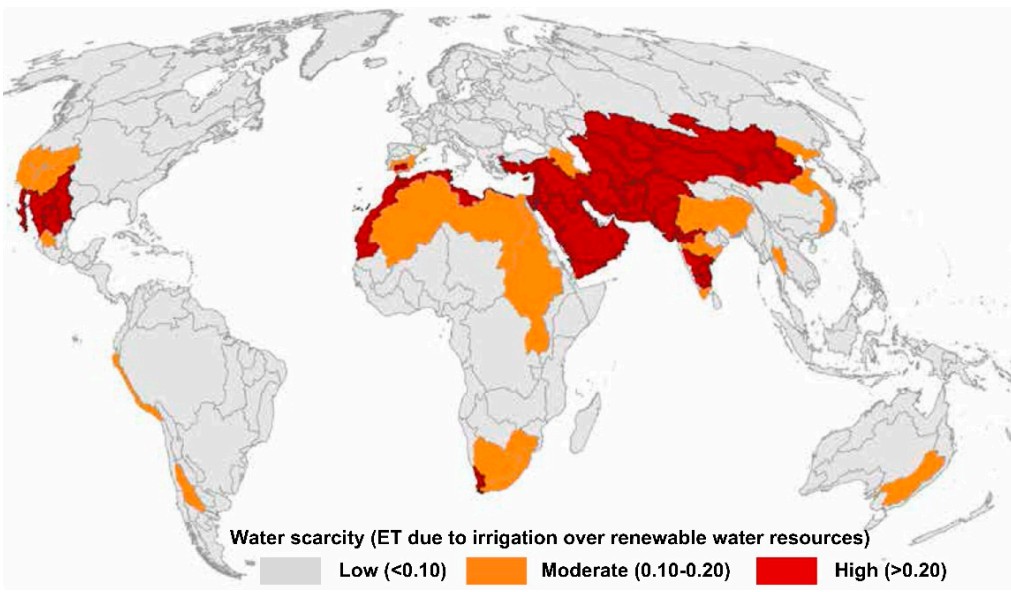

**Figure 1.** Global map of physical water scarcity by river basin [1].

In addition, global climate change is expected to further negatively affect water supply, crop production and thus food security in Jordan and other water-scarce countries. Therefore, the aim of this article is to provide a comprehensive overview of the impacts of agricultural cropping patterns and water security in one of the world's most water-scarce countries, Jordan, based on the most recent and relevant national databases. We believe that this study will contribute to more sustainable water and agricultural cropping pattern planning, climate change mitigation and national policy development to strategize the assistive scheme and adaption measures in Jordan.

## 2. Methodology

This research was initiated after the Jordan's Water Minister declared that 50 million $m^3$ lack in drinking water will be faced in 2022. This research is based on systematic approach Figure 2; where the problem of interest was clarified, then the governing factors were analyzed based on Market Data and the official records of the last 20 years; the country's strategy data for water were analyzed to identify the water share among the different sectors and the available distortions. Water-consuming sectors, including agriculture, were analyzed based on data from Jordan central market of vegetables for production amounts and prices. Official records from the Department of Statistics, FAO and Ministry of Water and Irrigation (MWI) were integrated to locate distortions of food and water in the country. Recent scientific research papers related to water and food scarcity were studied and analyzed to address scientific solutions assisting policy makers in making the right decisions. Jordan climatic data were used to solve insufficient cereals crops in Jordan, as well as to point out the potential land suitable for cereal planting. This research aims to address results assisting policy makers and contribute to more sustainable water and agricultural cropping pattern planning.

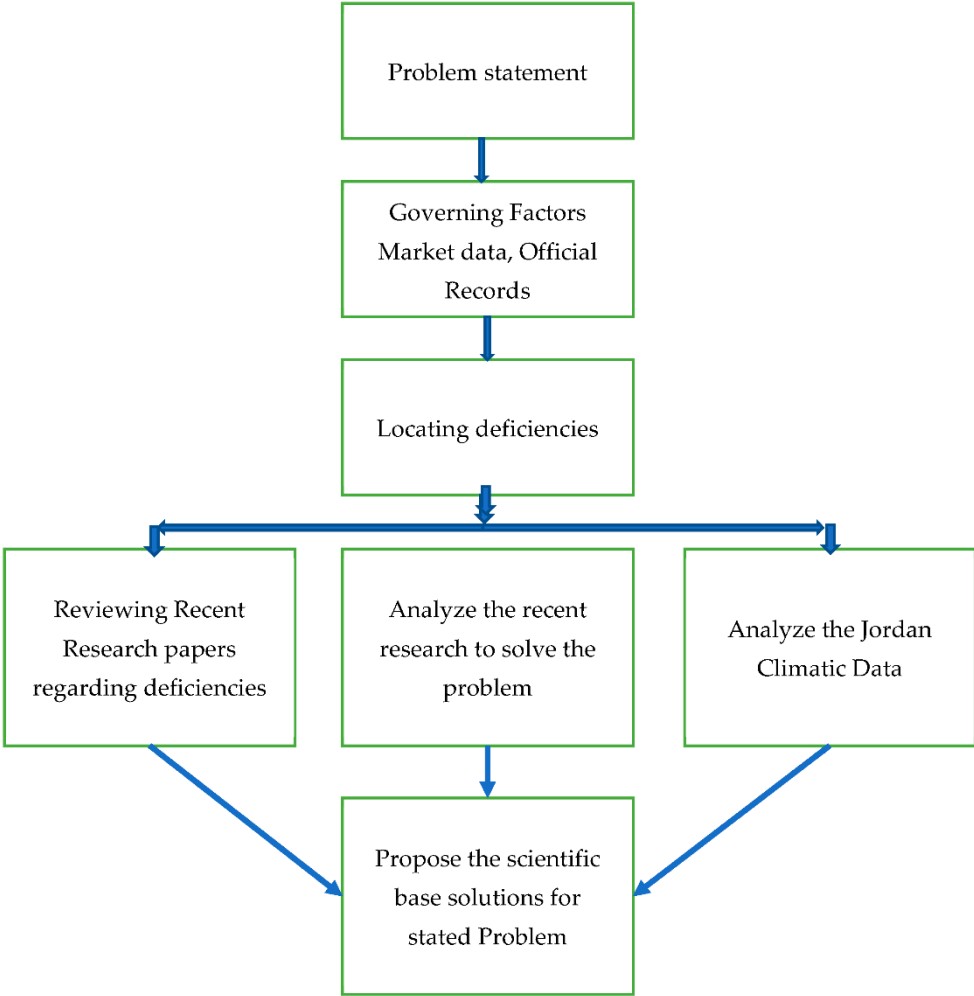

**Figure 2.** Methodological framework for the research performed.

### 3. Jordan, the Middle Eastern Country

Jordan is a Middle Eastern country characterized by low water resources (i.e., high water scarcity, Figure 1), and located in a politically conflicted area [5]. Collaborations with the surrounding countries regarding food and water are always in doubt. Jordan had a population about 10.81 million in 2020, 31% of which were refugees from Syria, Iraq and other countries [6]. This high refugee percentage creates stress in all sectors in Jordan, including economic, education, food and water sectors. According to the High Commissioner for Refugees in Jordan, there are 702,506 refugees and 4870 asylum seekers in Jordan (2020), and 90% of them live in Jordanian towns and cities and not in camps [7].

Authors analyzed the change in food prices in Jordan. The increase in the market food prices in 2013 is shown in Figure 3; it belongs to the sudden increase in Jordan's population (including all refugees), reduction of available water resources and the type of planting used in Jordan. As the political issues are difficult to control, Jordan should work on stabilizing the food prices and quantities in the markets in parallel with saving and managing the used water resources in more sustainable way. A noteworthy relationship exists between the water possessions of a country and the capacity for food production; assessing the irrigation needs is indispensable for water resource planning in order to meet food needs and avoid excessive water consumption [8]. To be able to control the food quantities and food prices, a detailed study should be performed on the Agricultural Cropping Patterns (ACP) from a productivity and water consumption point of view.

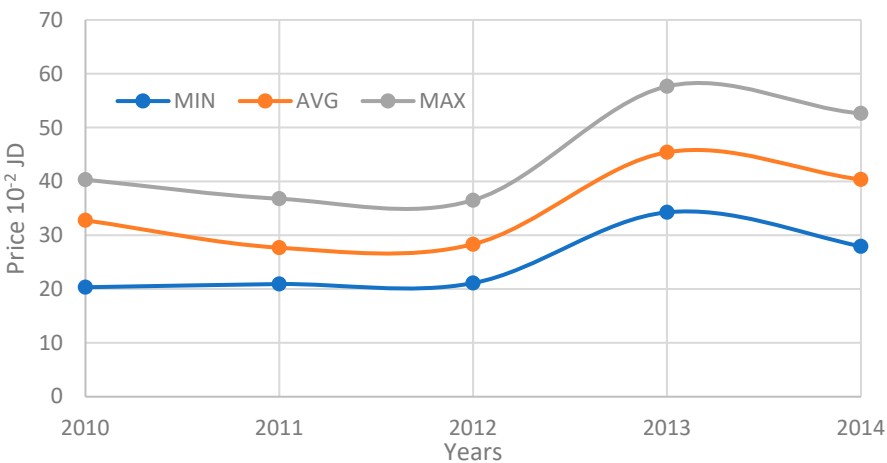

**Figure 3.** Prices jump as a result of Syrian immigration to Jordan.

## 4. Water Status in Jordan

Jordan area is 89,318 km$^2$, most of which is a semi-desert area [9]. Only 11.2% of the country is made up of plains, and 0.6% is heights, as shown in Figure 4. The average annual rainfall in Jordan is about 8.1 billion m$^3$ [10].

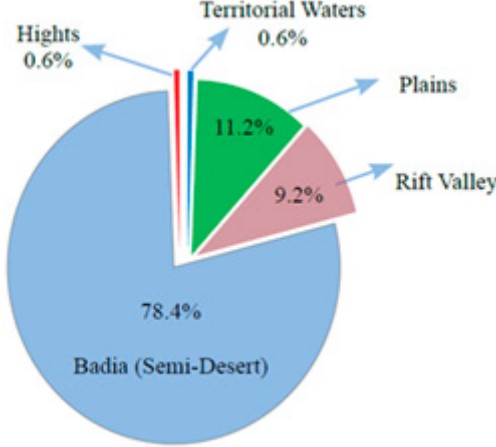

**Figure 4.** Kingdom of Jordan area by topography [11].

The total water consumption in Jordan reached 1053.6 Mm$^3$, with a dominant consumption in the agri-sector (52–60%), followed by municipal (40%) consumption. Jordanians receive about 78 m$^3$/person/year from renewable resources, while the worldwide amount per capita is 7000 m$^3$/person/year [12]. In 2021, Jordan's Water Minister declared that the country had about a 50 Mm$^3$ lack in drinking water. The network water leakage problem, which is estimated to be 52% of the supplied water, dramatically increased the water scarcity [13]. The wastewater treatment, which counts for 125 Mm$^3$/year, may be used for agriculture and does not solve the potable water problem.

The water scarcity in Jordan forced many farmers to over-exploit agricultural ecosystems in order to increase their production and incomes. This has caused a reduction or degradation of environmental sustainability, reduced farmers' profitability and led many producers to abandon rural areas [14,15]. To save water, the infrastructure of water distribution must be taken into consideration [16].

It is very clear that Jordan is suffering from a shortage of potable water and water used for irrigation. Network maintenance and local water harvesting in the high-rain-intensity areas could reduce the potable water problem for the coming years, and sea water

desalination using the available solar energy is a must in the near future. The irrigation and agriculture in Jordan will be discussed in the next few sections.

## 5. Agriculture in Jordan

Unsustainable agricultural cropping patterns (e.g., long-term mono-cropping) may be one of the main factors contributing to food insecurity. Different ACP are used worldwide depending on water availability in the specified area. Jordan, as a water-scarce country, pays a great deal of attention to the irrigation techniques in agriculture. Many researchers noted that water scarcity can be controlled in irrigation water management via proper choice of crops and farming patters [17]. Irrigation systems are very important from the perspective of water-use efficiency, which reaches 90% in topical systems, 80% in sprinklers and only 50–60% in surface irrigation [18]. Aquaponics units are promising solutions for small-scale cultivation, e.g., up to 3 $m^2$ for 1 $m^3$ of water [19]. Other options to reduce water consumption in planting may be to produce vegetables in tunnels, such as those in Jordan Valley (10 to 20%) [15]. Creating low water consumption plants of specific genetics takes a part of water conservation in agriculture; the pattern of genetic variability is considered as an important component of germplasm collection and conservation used in collaboration of crop's improvement process that includes the selection of parents for making new genetic recombination [20]. Irrigation management occupy a major role of water conservation (Conservation Agriculture) in conjunction with other complementary good agricultural practices of integrated crop and production management [21]. Many farmers used integrated crop livestock combination for optimum water usage [22–25]. Farmer-based and country-based solutions were proposed to improve water and food security [26]. Agricultural policies play a key role in sustainable agricultural development [3]. However, the challenge would remain the development in agricultural patterns and the efficient use of water resources as a means for future sustainable development [4].

### 5.1. Planted Area and Production

Food security is a measure of the crop's quantities produced to satisfy the needs of a community. The crop's production quantities depend mainly on the planted area. The total cultivated area in Jordan reached 0.27 million hectares, the irrigated area in Jordan was 0.11 million hectares (38%) and the rainfed area in Jordan is 0.17 million hectares (62%) of the total cultivated area. [27].

Table 1 indicates the reduction in the cultivated area through 2014–2019 due to the reduction of the available water. Not only Jordan is affected; the MENA region faces many challenges, including food and water security, and the region will be heavily affected by climate change as the century evolves [28]. If these issues are not addressed seriously, they will raise perilous development challenges for the entire region. The region needs innovative integrated approaches to provide opportunities to cap with food and water needs [29,30]. Financing, water needs, food security, poverty and nutrition in the MENA region have been discussed to highlight major features and challenges that face MENA countries to achieve self-sufficiency and food security [31–33]. A look at Jordan's food production and needs can indicate the self-sufficiency ratio (SSR) of each vegetable product. Table 2 shows that SSR in the year 2020 for different vegetables was ≥100% in almost all vegetables [34]. From this table, Jordan produces more than its needs from vegetables consuming more water. The extra water used in vegetables may be directed to other important crops.

**Table 1.** Cultivated area in Jordan [27].

|  | 2014<br>Hectare | 2019<br>Hectare |
| --- | --- | --- |
| Field crops | 138.6 | 110 |
| Vegetable crops | 50.9 | 33.4 |

**Table 2.** Self-sufficiency ratio (SSR) of vegetables (2020) [34].

| | 2020 | | | |
|---|---|---|---|---|
| | **SSR (%)** | **Exports (Ton)** | **Imports (Ton)** | **Production (Ton)** |
| Tomatoes | 146.9 | 184,300.10 | 0 | 577,287.90 |
| Cucumbers | 107.8 | 10,265.60 | 0 | 141,382.80 |
| Squash | 128.5 | 18,965.30 | 0 | 85,460.60 |
| Eggplants | 104.9 | 2691.20 | 0 | 57,355.70 |
| Cauliflower | 118.6 | 8831.80 | 87 | 55,685.80 |
| Cabbages | 107.3 | 3599.20 | 1.1 | 52,616.40 |
| Onion, dry | 103.8 | 3008.20 | 12.1 | 82,163.10 |
| Carrots | 90.2 | 47.6 | 1817.40 | 16,326.20 |
| Broad beans, green | 99.2 | 115.1 | 177.1 | 8046.00 |
| Watermelons | 119.3 | 14,134.00 | 0 | 87,498.80 |
| Sweet melons | 159.9 | 20,102.80 | 2.3 | 53,653.60 |

Figure 5 illustrates the planted area containing vegetables in the Jordan Valley, as the production in Amman and Al Mafraq is negligible. Most of the area under cultivation in Jordan is planted with tomatoes (Figure 5). Jordan is considered one of the source countries for tomatoes in the region, and exports almost 30% of its production to different countries [35]. In conclusion, Jordan has enough area planted with vegetables that covers its needs. Nevertheless, available agricultural land should be carefully used to satisfy the food security in all types of foods, not only vegetables, as will be discussed later in this work.

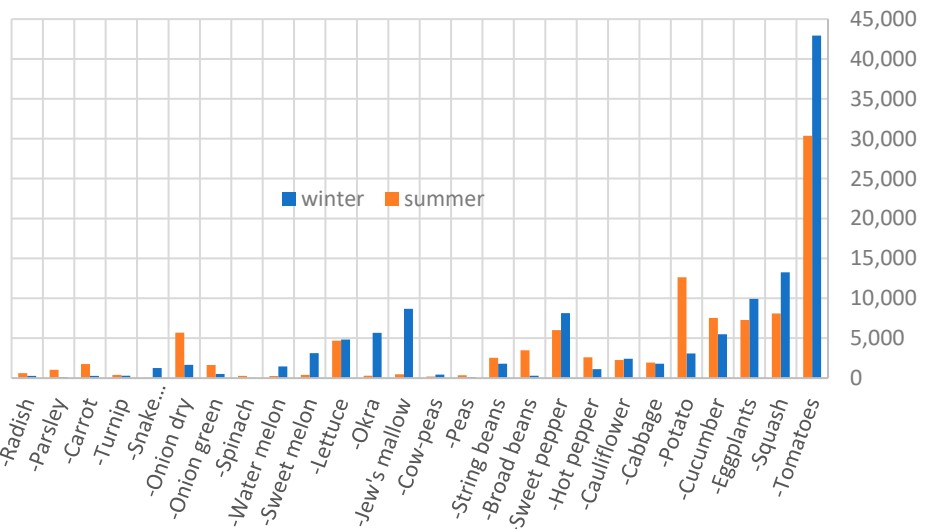

**Figure 5.** Vegetables planted area in summer and winter (Jordan Valley) [34].

### 5.2. Water Consumption and Irrigation Systems

Vegetable plants' need for water varies from one crop to another [36]. Some vegetables are tolerant of a lack of irrigation water; this factor is controlled by the distribution of plant roots [37,38]. It is noted that plants with deep roots (tomatoes, pumpkins, sweet potatoes, etc.) consume water in smaller quantities than plants with surface roots (potatoes, onions, cabbage, broccoli, zucchini, garlic, lettuce, parsley, spinach, radish, etc.) [39–41]. In general, the production of vegetables requires large quantities of water. On average, vegetables of all sorts need about 350 l/m$^2$ for one crop cycle [42,43]. For one dunum or 1000 m$^2$ (decare), a total of 350 m$^3$ water is needed [44]. Pakistan's irrigation system is one of the most inefficient irrigation systems, where more than 60% water is lost during conveyance in the channels and application in the field [45]. Researchers highlighted the importance of water for food security, they required allowance of water prices to project their real cost

and ensure the financial sustainability of water utilities [46]. Other researchers suggested new strategies for water development and management to increase water supply; they emphasize increasing the investment in the irrigation sector to improve irrigation use efficiency [47]. To improve the irrigation efficiency, the transition from flood irrigation to subsurface irrigation or sprinkler and drip irrigation is a must [48,49]. Table 3 shows the cultivated area percentage with different irrigation systems used. It is clear that most vegetables in Jordan are using a drip irrigation system. Drip irrigation systems reduce labor requirements by half, and double the planted area with the same amount of water. The low-cost drip system is likely to be adopted by small farmers in semi-arid and hilly regions [50].

**Table 3.** Cultivated area percentage of vegetables by type of crop, plantation and irrigation system (DOS, 2017).

| Crop | Plastic Houses % | | Plastic Tunnels % | | Open Field% | | | Drip Irrigation % |
|---|---|---|---|---|---|---|---|---|
| | Drip | Surface | Drip | Surface | Drip | Surface | Sprinklers | |
| Tomato | 25.4 | 0.6 | 2.14% | 0 | 68.83 | 2.67 | 0 | 96.4 |
| Squash | 2.23 | 0.1 | 7.4 | 0 | 88 | 1.1 | 0 | 97.63 |
| Cucumber | 95.3 | 1 | 0 | 0 | 1.6 | 1.6 | 0 | 96.9 |

High-technology drip irrigation systems [51] are dominantly used in Jordan, either in greenhouses or in open field environments.

Fortunately, there are already ways to dramatically reduce the amount of water needed in agriculture. Advanced artificial intelligence models and deep reinforcement learning may be combined to make smart decisions related to the amount of water needed for crop growth [52]. Vegetables take about 60 L of water to produce a single kilogram of product if they are grown outdoors [53]. This falls to 15 L for hydroponic greenhouse cultivation, and then to just 4 L per kilogram when the very latest indoor growing technology is used [54–57].

The water-use efficiency (WUE) and the product water use (PWU) are terms to evaluate the kg of product produced per liters of water used and used water liters per kilogram of product produced, respectively [58]. The water consumption in Jordan is still high (about 300 L/kg of fresh vegetable) due to the agriculture pattern used (open field). Open systems used for agriculture suffer from evaporation from the open surface, transpiration by leaves and drain from the root zone [59]. There will be a large improvement created by moving from an open field drip irrigation system to a greenhouse drip irrigation system, which will increase the water-use efficiency, increase production and decrease transpiration [60]. Such transformations in agriculture in Jordan can reduce water consumption as much as five times compared to the open field type (about 60 L/kg fresh vegetables). Further modifications can be achieved by moving to controlled greenhouse-type agriculture where the temperature and radiation is controlled within the greenhouse; again, drip irrigation is used to save water usage [61]. In such cases, a further drop in water usage up to four times can be achieved; the water consumption per kg may drop down to 15 L per kg fresh vegetables.

Planting with sand and other substrates was tested to reduce the water consumption and increase product [62]. Water scarcity and the need to produce adequate food for the increasing population leads to the search for technological solutions that allow crops to be grown with less water. High-tech solutions in greenhouse production lead the way to water efficient production techniques such as hydroponics [63]. This highly cost-effective solution makes it possible to produce quality fruit and vegetables without depending on soil conditions [64]. An improved greenhouse type can lead to higher production. Examples of an improved greenhouse are found at the premises of Bakker Brothers (a seed company uses greenhouses for vegetables plantation) in Amman. This type has rooftop ventilation,

which can provide a better climate inside the greenhouse, and is higher compared to the present tunnels used [65]. Moreover, the pricing policy of water is considered to be an effective way to reduce water consumption and to use water in a more efficient manner, as mentioned [44]. In water-scarce countries, more investments should be directed toward the closed greenhouses high-technology hydroponic system, which reduces the amount of water consumption almost three times compared to conventionally controlled green houses.

### 5.3. Crops Price Fluctuation and Stability

As mentioned above, in Jordan, most plantations use a drip irrigation open field system. Such systems are affected greatly by sun light and temperature. Figures 6 and 7 show the average onion and tomato production and prices all over the year, respectively. They indicate the high fluctuation in product supply to the central market. The prices of the products are also highly fluctuating with respect to average prices. Figures 6 and 7 suggest reducing the production to offer a more suitable price and save water for the strategic crops, such as wheat and barley.

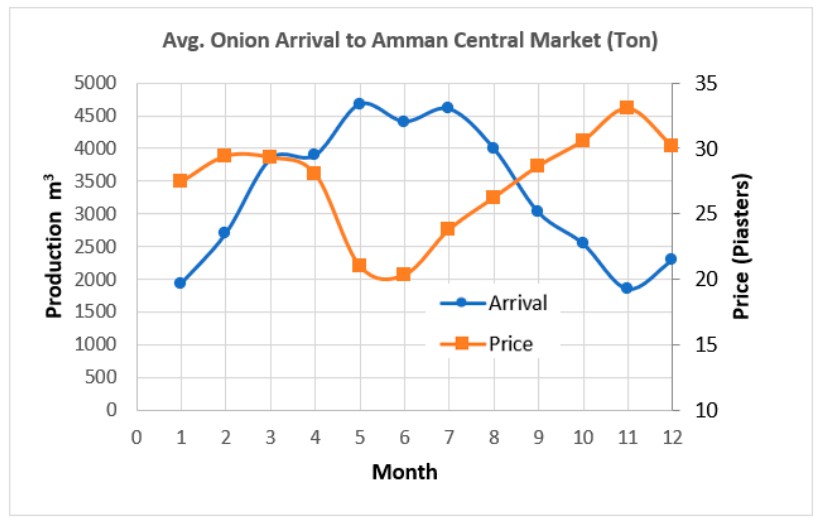

**Figure 6.** Average onion production and prices in Amman central market throughout the year.

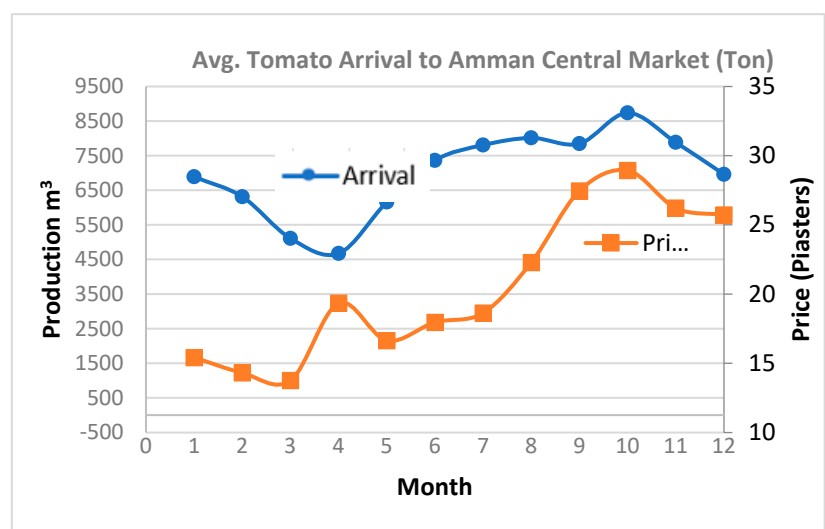

**Figure 7.** Average tomato production and prices in Amman central market throughout the year.

### 5.4. Strategic Crops in Jordan

On the other hand, Jordan is completely dependent on outside sources for its cereal consumption, as shown in Table 4. Although wheat and barley are strategic crops [34,66], the government attributes the lack of cereal production to the water shortage from which Jordan suffers.

**Table 4.** Self-sufficiency ratio of cereal (2020) [34].

| | 2020 | | | |
|---|---|---|---|---|
| | SSR (%) | Exports (Ton) | Imports (Ton) | Production (Ton) |
| Wheat | 2.5 | 1390.70 | 968,489.10 | 24,360.80 |
| Rice | 0 | 2146.30 | 213,884.20 | 0 |
| Barley | 12 | 6696.40 | 617,825.50 | 83,598.30 |
| Maize | 0 | 9488.00 | 805,907.80 | 0 |
| Millet | 0 | 0 | 1281.20 | 0 |

Referring to Table 2, calculating the consumed water amounts needed to produce over 100% SSR products (such as tomato, squash and sweet melon), about 80 $Mm^3$ is saved. This sum of available water can be directed to produce strategic crops for people. Actually, water-scarce countries such as Jordan do not have the luxury to use water anywhere. Water resources should be controllable and used in modern closed high-technology-controlled greenhouse agriculture. Currently, 512.146 $Mm^3$ of water is used for agriculture [67]; a simple calculation can be performed according to [39] to find the amount needed to produce our vegetables in moderately controlled greenhouses. The calculations yield a 346 $Mm^3$ saving in water resources.

Table 4 shows that Jordan is not self-sufficient in cereals, importing almost 96% of its needs. Wheat and barley are considered among the main strategic crops for the country. To achieve food security, country should concentrate on wheat and barley planting to ensure self-sufficiency. Even though planting cereals consumes a lot of water [68], moving to high-tech agricultural systems may allow the country to cover its needs. Many studies have been carried out on wheat planting using high-tech systems [69,70]. Wheat, barley and fodder hydroponic were tested and are still under investigation; good amounts of fodder were produced using hydroponic systems [71–73]. Cereals may be grown hydroponically [74,75], or in assisted open fields. Figure 8 shows the predicted maximum average temperatures and amounts of rainfall in Jordan as zones. It is clear that Jordan has a variety in temperatures and rainfall. Wheat and barley require a temperature of about 18 °C and, in terms of rainfall, about 31 mm/month [76]. Analyzing Figures 8 and 9, a large area from north Jordan to Shoubak in the midsouth is suitable for planting cereal in open field without extra water consumption. Such a plantation in the far ends of the suitable zones may require assistance with irrigation.

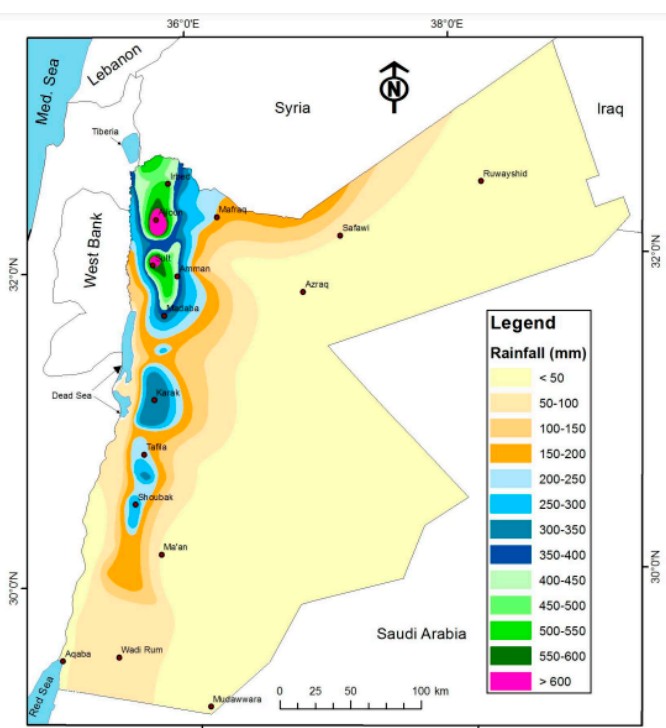

**Figure 8.** Rainfall distribution in Jordan [14].

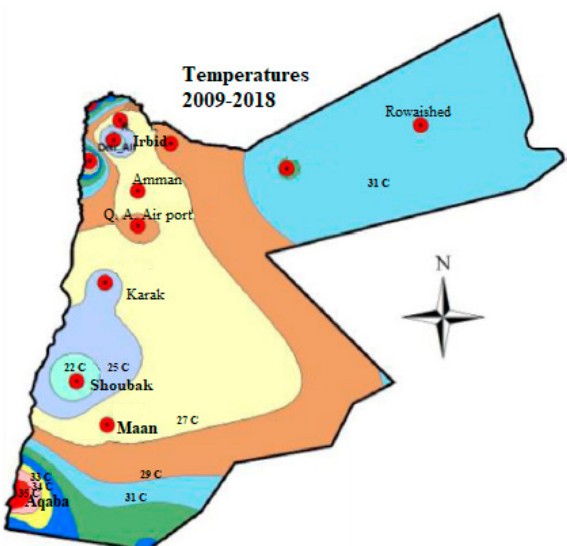

**Figure 9.** Maximum average temperature zones in Jordan.

## 6. Conclusions

Jordan is one of the most water-scarce countries, where an active migration process and aging infrastructure in the municipal water supply network put additional pressure on water and other natural resources. For example, modernizing the municipal water supply system could save nearly 50% of total water consumption in this country. In addition, Jordan could reduce water consumption by five-fold if agricultural production was transferred from open-field to greenhouse cultivation, but this option requires huge infrastructural investments. In addition, it has been shown that more investments should support closed (with nutrient recycling) hydroponic greenhouses production, which use water and nutrients more efficiently, e.g., up to three-fold greater efficiency compared to conventional open greenhouses (without recycling). It is recommended that after achieving self-sufficiency in horticultural crops (e.g., tomatoes, pumpkins, sweet melons), water

resources from the open field cropping system should be transferred (e.g., from northern Jordan to Shoubak in the Central–South region), where cultivation is possible with almost no additional water use, which will further contribute to alleviating food insecurity in Jordan.

**Author Contributions:** N.B. conceptual and draft writing, A.Q. and M.B.K. review and editing, J.H. and G.O. funding and review. All authors have read and agreed to the published version of the manuscript.

**Institutional Review Board Statement:** Not applicable.

**Informed Consent Statement:** Not applicable.

**Data Availability Statement:** Not applicable.

**Acknowledgments:** Authors are grateful to the Applied Science Private University, Amman, Jordan, for the financial support granted to this research. Authors are grateful to the Tafila Technical University, Tafila, Jordan, for the financial support granted to this research. Authors are grateful to the Zaytoonah University of Jordan, Amman, Jordan, for the financial support granted to this research.

**Conflicts of Interest:** The authors declare no conflict of interest.

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
