# Peer review of "Review of Agricultural-Related Water Security in Water-Scarce Countries: Jordan Case Study"

_agronomy, doi:10.3390/agronomy12071643_

Round 1

Reviewer 1 Report

The topic is interesting but the paper needs more than a major revision  before it can be published.

In the following you find some comments to improve the paper.

Please point out the novelty of this research. This is not really clear.

The structure of the paper is not common. There is missing the methodology section. Please explain which methodology is used? What is the database?

Comments regarding the abstract:

  • 78m³/person: is this value correct? and there is missing a relation: per year? per day?
  • 50 Mm3 : why here other units?

Conclusion: Is greenhouse planting the best solution? This results in additional land sealing. What effects does that have?

Legend in the maps is mostly not really readable. Please insert and check the units (rainfall mm?)

Reviewer 2 Report

As shown in the attachment.

Author Response

I would like to thank Reviewer (2) and here are the reply for his comments,

Major

1- Title of the article: The term 'the Agricultural Patterns (AP)'
Reply: the title has been changed according to reviewer advice....

Minor

1-P2L42-43 Ê»Jordan is recognized as the second-most water stressed country in the worldʼ 

given reference [13]
2-P2L66-67: explanation in Fig.2; Ê»the increase in the market food prices is clear in Figure 2ʼ- The spike in food prices appears to have occurred in 2013, which should be clearly stated in the text. 

Reply: the year 2013 has been stated in the text.

3-P5L147-148: explanation in Table 2
- What can be said from this table should be described.

Reply: the following explanation been added

"Table 2. shows that SSR in the year 2020 for different vegetables was > 100% in almost all vegetables [34], from this table Jordan produces more than its needs from vegetables; consuming more water. The extra water used in vegetables may be directed for other important crops."

‏4-P9-10: Figures 5 and 6
-For the blue line, its legend should also indicate that it is Arrival. Also, since these two
figures are in the same format, they should be combined as a new Figure 5

Reply: Legend added for blue line, but it will be too small to combine both figures in one figure... (Arrival amounts are very different).

Round 2

Reviewer 1 Report

Thank you for the revision. But there are still some points missing:

Please explain how it stands out from other research. In my opinion, only the objective of the research was better forganized but not the novelty of the research approach. Here the work still needs to be optimized.

Also, the methodology addressed is still not clearly formulated. The methodology should be embedded in a separate chapter not at the end of the introduction section.

Author Response

This manuscript is a resubmission of an earlier submission. The following is a list of the peer review reports and author responses from that submission.

Round 1

Reviewer 1 Report

As shown in the attached file.

Reviewer 2 Report

In general, the article is dealing with an interesting topic. But before it can be published more background information and a deeper research is necessary. The methodolgy has to be clearly described and pointed out. Also the novelty of this research has to be pointed out.

Here are some specific comments:

Abstract: This sentence is not clear for me: While the world-wide per capita share is 7000 meters cube per person, and water stress is 1000 m3/person, Jordanian gets about 78 m3/person from renewable resources. Please be more specific. Why is the unit meters cube and not m³? Then the unit Mm³? Why is this not consistent?

Is there a reference for figure 5 or is it self-analyzed? Here is more background information necessary. What is the data base? Is it analyzed by aerial photos?

Figure 9 is part of the case-study description. Why is this presented in the chapter where normally the results are described?

The part (chapter 4) on costs is described very briefly. Why are only certain crops discussed here?

Regarding the conclusion: Is greenhouse planting the solution? This results in additional land sealing. What effects does that have? You can't look at it in an entirely positive light.